# Comparative Serum Proteome Profiling of Canine Benign Prostatic Hyperplasia before and after Castration

**DOI:** 10.3390/ani13243853

**Published:** 2023-12-14

**Authors:** Sekkarin Ploypetch, Grisnarong Wongbandue, Sittiruk Roytrakul, Narumon Phaonakrop, Nawarus Prapaiwan

**Affiliations:** 1Department of Clinical Sciences and Public Health, Faculty of Veterinary Science, Mahidol University, Nakhon Pathom 73170, Thailand; sekkarin.plo@mahidol.ac.th (S.P.); grisnarong.won@mahidol.edu (G.W.); 2Functional Proteomics Technology Laboratory, National Center for Genetic Engineering and Biotechnology, National Science and Technology Development Agency, Pathum Thani 12120, Thailand; sittiruk@biotec.or.th (S.R.); narumon.pha@biotec.or.th (N.P.)

**Keywords:** benign prostatic hyperplasia, castration, dogs, proteome, serum proteins

## Abstract

**Simple Summary:**

Benign prostatic hyperplasia (BPH) is a common condition in older intact dogs, characterized by nonmalignant growth of the prostate gland. However, diagnosis and treatment strategies vary widely. Clinical proteomics is a rapidly growing field in both human and veterinary medicine. It has the potential to help us discover new therapeutic targets and biomarkers for diagnosis, prognosis, and treatment efficacy. However, there has been a notable gap in our understanding regarding changes in the serum proteome of dogs with BPH. This study aimed to compare serum proteome profiles of dogs with BPH to healthy dogs and evaluate the impact of castration. This research represents the first investigation into serum proteome profiling in dogs with BPH before and after castration. The findings suggest that specific serum proteins may be associated with the development and progression of canine BPH. These proteins hold the potential to serve as non-invasive biomarkers for diagnosing the disease. Further study is needed to identify and validate diagnostic biomarkers for BPH in dogs, including proteins involved in related pathways, as numerous alterations in serum proteins were observed post-castration.

**Abstract:**

BPH is the most prevalent prostatic condition in aging dogs. Nevertheless, clinical diagnosis and management remain inconsistent. This study employed in-solution digestion coupled with nano-liquid chromatography tandem mass spectrometry to assess serum proteome profiling of dogs with BPH and those dogs after castration. Male dogs were divided into two groups; control and BPH groups. In the BPH group, each dog was evaluated at two time points: Day 0 (BF subgroup) and Day 30 after castration (AT subgroup). In the BF subgroup, three proteins were significantly upregulated and associated with dihydrotestosterone: solute carrier family 5 member 5, tyrosine-protein kinase, and FRAT regulator of WNT signaling pathway 1. Additionally, the overexpression of polymeric immunoglobulin receptors in the BF subgroup hints at its potential as a novel protein linked to the BPH development process. Conversely, alpha-1-B glycoprotein (A1BG) displayed significant downregulation in the BF subgroup, suggesting A1BG’s potential as a predictive protein for canine BPH. Finasteride was associated with increased proteins in the AT subgroup, including apolipoprotein C-I, apolipoprotein E, apolipoprotein A-II, TAO kinase 1, DnaJ homolog subfamily C member 16, PH domain and leucine-rich repeat protein phosphatase 1, neuregulin 1, and pseudopodium enriched atypical kinase 1. In conclusion, this pilot study highlighted alterations in various serum proteins in canine BPH, reflecting different pathological changes occurring in this condition. These proteins could be a source of potential non-invasive biomarkers for diagnosing this disease.

## 1. Introduction

Benign prostatic hyperplasia (BPH), prostatitis, prostatic cysts, prostatic abscesses, and fewer cases of neoplasia like squamous metaplasia are prevalent prostatic diseases in male dogs [1]. BPH is the most common prostatic disease that affects intact male dogs and develops spontaneously as glandular hyperplasia [2]. It is mostly the consequence of aging and increasing dihydrotestosterone (DHT) levels in intact male dogs, which causes an increase in the size (hypertrophy) and number (hyperplasia: the main mechanism in dogs) of the prostate epithelial cells [1,3]. DHT, an androgen derivate that collaborates with its receptors on the prostate, is a crucial hormone in the pathophysiology of this condition, enhancing the growth of both the stromal and glandular components of the prostate, which leads to its enlargement [4]. Growth factors such as insulin-like growth factor 1, epidermal growth factor (EGF), fibroblast growth factor, and vascular endothelial growth factor (VEGF) are activated when DHT binds to androgen receptors, which may cause prostatic enlargement and the development of BPH [5,6]. Estrogens increase the number of androgen receptors and promote metaplasia, which is a change in the structure of prostatic cells [7]. Moreover, the research on the potential mechanism of inducing BPH in dogs by estradiol suggested that estrogens undergo conversion into catechol estrogen, a form of estrogen metabolites. These metabolites, produced through this conversion process, then actively participate in metabolic redox cycling—a process that generates free radicals. Within this intricate pathway, cytochrome P450 enzymes play a crucial role by oxidizing catechol estrogens to quinones via semiquinone intermediates. The formation of semiquinones or other free radicals through metabolic redox cycling has the potential to cause damage to cellular macromolecules. When these free radicals impact prostatic cells, they may respond to DHT-mediated growth stimuli, ultimately leading to the development of BPH [8]. Prolactin, produced by the anterior pituitary gland, may potentially contribute to prostatic hyperplasia by enhancing the epithelial stroma via a mechanism that controls the expression of granulocyte-macrophage colony-stimulating factor via sustained signal transducer and activators [9]. Patient history, clinical signs, physical examination, rectal palpation of the prostatic contour, radiographic measurement of the prostatic size, ultrasonographic measurement of the prostatic volume and parenchyma, ultrasound-guided fine needle aspiration, and excisional biopsy are all methods used to diagnose BPH [10]. However, ultrasonography is the diagnostic method of choice for examining the prostate, allowing for assessing both the gland’s size and the homogeneity of its parenchyma [11]. The length, width, and depth of the prostate gland and its internal structure and exterior texture can all be examined using ultrasound [10]. In intact male dogs, age and body weight also have a positive correlation with the prostate’s size [12]. Due to the similarities in clinical and ultrasonographic findings, BPH can be challenging to distinguish from other prostatic diseases, such as prostatitis, prostatic cysts or abscesses, and prostatic neoplasia [7]. However, using ultrasound-guided FNA for diagnostic purposes in canine prostatic diseases requires extreme caution due to concerns about needle-track seeding with infectious agents and neoplastic cells [13]. Furthermore, some researchers agree that excisional biopsy, as an invasive operation, involves a risk of consequences such as infection, which can lead to inflammation of the bladder, prostate, testicles, and spermatic duct [14]. Various tests for specific biomarkers, including prostate-specific antigen (PSA), Kallikrein 2 (hK2), urokinase plasminogen activator (uPA), transforming growth factor-beta 1 (TGF-Beta 1), interleukin-6 (IL-6), VEGF, prostate cancer antigen (PCA), CCL11/Eotaxin chemokine, and also microRNA (miRNA) are performed in the diagnosis of prostate diseases in men as an addition to the standard clinical examination. According to these studies, biomarker identification is common, practical, and minimally invasive, and it should be recognized that they have excellent prognostic and diagnostic value [15,16,17,18]. There is currently little information available in the literature about the use of biomarkers in canine prostate diseases. Canine prostate-specific esterase (CPSE), prostatic acid phosphatase (PAP), and PSA are some biomarkers of the male canine reproductive tract that may be assessed for the diagnosis of BPH [19]. A recent study assumed that miRNA-129 and VEGF determination may be useful for diagnosing BPH in dogs [20]. There is currently an absence of studies on the use of serum biomarkers for canine prostate diseases, particularly BPH.

Although no treatment is suggested for dogs with subclinical and mild signs of BPH, some treatment should be recommended when symptoms affect the quality of life [21]. In breeding dogs, treatment with pharmacological compounds inhibiting the production or activity of androgens such as 5α-reductase inhibitors (finasteride) and α1A-adrenergic receptor antagonists may be preferred to maintain fertility [22]. Castration, which permanently removes the androgenic stimulation of BPH, decreases the serum concentrations of testosterone and estrogen, and results in a decrease in the size of the prostate, is known as the gold standard treatment for BPH, especially in dogs not intended for breeding [23]. Up to 70% of the prostate’s size is decreased by castration [10]. A current study proposed that castration is an effective and long-lasting treatment for dogs with BPH, as it causes a rapid decrease in serum testosterone concentrations (within 7 days) and a decrease in glandular volume, similar to the volume of healthy animals (within 14 days) [24]. Following castration, the early phase of the decrease in prostatic volume was accompanied by an immediate decrease in circulating testosterone levels; moreover, prostate atrophy was encouraged, which altered the prostatic artery’s hemodynamics [25,26].

Clinical proteomics, a rapidly developing field with increasing significance in humans as well as in veterinary medicine, shows promise for the discovery of novel therapeutic targets and biomarkers for diagnosis, prognosis, and therapeutic efficacy by utilizing current technology to compare proteome profiles between various physiological and disease states [27,28]. Studies on proteomics in canine reproductive problems such as mammary tumors [29] and pyometra [30] have been published, though they are less numerous than those in humans. To the authors’ best knowledge, however, the possible changes in serum proteome in canine BPH have not yet been studied. This study aimed to compare dogs with BPH to healthy dogs and between dogs before and after castration to assess potential alterations in serum proteome profiling.

## 2. Materials and Methods

### 2.1. Animals and Sample Collection

Thirty-five client-owned male dogs undergoing castration at Prasu-Arthorn Animal Hospital, Faculty of Veterinary Science, Mahidol University were classified into healthy dogs and dogs diagnosed with benign prostatic hyperplasia. Two main groups of dogs were identified and categorized as the control group and the benign prostatic hyperplasia group (BPH group). The classification process was performed based on several assessments, such as patient history, clinical examination, digital rectal palpation of the prostate, complete blood count (CBC), and serum biochemical analysis. Dogs in the control group must be in the puberty age range (1–7 years), healthy with no chronic illnesses, and exhibit normal results in digital rectal prostate palpation and blood tests. Twenty healthy dogs, aged from 1 to 4 years, of different body weights (4 to 30 kg) served as a control group (CTRL). Dog breeds included mongrels (*n* = 8), the Pomeranian (*n* = 3), Chihuahua (*n* = 2), French Bulldog (*n* = 2), Siberian Husky (*n* = 2), Japanese Spitz (*n* = 1), Pit Bull Terrier (*n* = 1), and Shih Tzu (*n* = 1). The BPH group included fifteen dogs that were diagnosed with benign prostatic hyperplasia (Table 1). From the medical history screening, dogs in the BPH group have no pre-existing conditions such as cardiovascular disease, urologic disorders, gastrointestinal disorders, cancer, or endocrine diseases, with particular emphasis on reproductive system disorders, including cryptorchidism, orchitis, penis tumors, and testicular cancer. Moreover, these dogs, showing one or more clinical signs—sanguineous discharge from the prepuce or urethra, hematuria, tenesmus, or straining during defecation—were qualified [31]. Digital rectal examination revealed normal consistency and no pain. Blood tests showed no abnormalities. Additionally, ultrasonography was used to confirm BPH based on homogeneity of the parenchyma and gland size criteria. The prostatic ultrasonographic assessment was performed transabdominally in each dog of the BPH group with a linear transducer (GE LOGIQ P6, GE HealthCare, United States). Before transabdominal scanning, the dogs’ caudal abdominal area was shaved and they were placed in a dorsal recumbent position. Four thematic categories were used with precise and descriptive terminology to identify the appearance of B-mode prostate parenchyma. These categories were background echotexture (normal, hyperechoic, or hypoechoic); parenchymal stippling (regular, increased, or coarse); general appearance (homo- and heterogenous); and focal changes (cysts, mineralized opacities, or focal hypoechoic lesions) [32]. In this study, dogs being considered for classification into the BPH group had to exhibit specific characteristics, including increased echogenicity or hyperechoic background echotexture, parenchymal stippling that may be coarse or regular, a general appearance that can be either heterogeneous or homogeneous, and focal changes such as small cysts measuring less than 0.5 cm or the presence of mineralized opacities [31]. Therefore, dogs with ultrasonographic findings of hypoechoic or anechoic cavities larger than 0.5 cm within prostate parenchyma, indicating prostatic cyst or abscess, were excluded. The ultrasonographic examination enabled the calculation of the prostate size using length, width, and diameter (dorsoventral distance). Prostatic volume (PV) was determined by the formula: PV (cm^3^) = (width (cm) × length (cm) × diameter (cm))/2.6 + 1.8; the estimated volume (EV) was determined by the formula EV (cm^3^) = 0.33 × body weight (kg) + 3.28 [19,33]. Each dog in this group was evaluated by ultrasonography at two time points: Day 0 (considered the day of BPH diagnosis; BF subgroup) and Day 30 (one month after castration; AT subgroup).

The 3 mL of blood samples were collected via the cephalic vein of all dogs before castration. One month later, blood samples from the dogs in the BPH group were collected again. Serum was obtained by centrifugation of blood collected in plain tubes. Aliquoted samples were stored at −20 °C until analyzed. Lowry’s assay evaluated total protein concentrations from serum samples using bovine serum albumin as a standard.

### 2.2. In-Solution Digestion by Trypsin

In each sample, 5 µg of proteins were reduced using 5 mM dithiothreitol (DTT) in 10 mM ammonium bicarbonate (NH_4_HCO_3_) at 60 °C for 1 h. Alkylation of the reduced cysteine residues was achieved by adding 15 mM iodoacetamide (IAA) in 10 mM ammonium bicarbonate (NH_4_HCO_3_) and incubating the sample in darkness at 25 °C for 45 min. Afterward, the protease trypsin was added at a ratio of 1: 20 (enzyme/protein) and allowed to act for 16 h at 37 °C. Finally, the samples were dissolved using 0.1% formic acid (FA) and assessed through nano-liquid chromatography tandem mass spectrometry (nanoLC-MS/MS) analysis.

### 2.3. LC-MS/MS

The tryptic peptide samples were performed for injection into an Ultimate3000 Nano/Capillary LC System (manufactured by Thermo Scientific, CHS, Chelmsford, UK) connected to a ZenoTOF 7600 mass spectrometer (produced by SCIEX, located in Framingham, MA, USA). In brief, one microliter of peptide digests was enriched using a µ-Precolumn (300 µm i.d. × 5 mm) packed with C18 Pepmap 100 (5 µm, 100 A, from Thermo Scientific, CHS, UK) and then separated on a 75 μm I.D. × 15 cm column packed with Acclaim PepMap RSLC C18 (2 μm, 100 Å) using nanoViper technology (also from Thermo Scientific, CHS, UK). The C18 column was enclosed within a temperature-controlled column oven set to 60 °C. Solvents A and B, containing 0.1% formic acid in water and 0.1% formic acid in 80% acetonitrile, respectively, were used for the analytical column. A gradient ranging from 5% to 55% of solvent B was employed to elute the peptides, maintaining a constant flow rate of 0.30 μL/min for 30 min.

For the ZenoTOF 7600 system, the source and gas parameters were configured as follows: ion source gas 1 was set to 8 psi, curtain gas to 35 psi, CAD gas to 7 psi, source temperature to 200 °C, polarity to positive, and spray voltage to 3300 V. Regarding the DDA method selection, it involved the selection of the top 50 precursor ions with the highest abundance from the survey MS1 scans for subsequent MS/MS analysis, with an intensity threshold set above 150 cps. Precursor ions were dynamically excluded for 12 s after two instances of MS/MS sampling (with dynamic CE for MS/MS enabled). MS2 spectra were acquired in the range of 100–1800 *m*/*z*, each with a 50 ms accumulation time, and the Zeno trap was enabled. The collision energy parameters included a declustering potential of 80 V, 0 V DP spread, and a CE spread of 0 V. The time bins to sum were set to 8 with all channels enabled, and a Zeno trap threshold of 150,000 cps was applied. The cycle time for the Top 60 DDA method was set to 3.0 s.

### 2.4. Bioinformatics and Data Analysis

MaxQuant version 2.2.0.0 was employed for protein quantification within individual samples, utilizing the Andromeda search engine to align MS/MS spectra with the Uniprot *Canis familiaris* database [34]. Label-free quantification was carried out using MaxQuant’s standard parameters, which included allowing a maximum of two missed cleavages, a mass tolerance of 0.6 daltons for the primary search, trypsin as the digestion enzyme, carbamidomethylation of cysteine as a fixed modification, and the consideration of methionine oxidation and protein N-terminus acetylation as variable modifications.

For protein identification, only peptides containing a minimum of 7 amino acids as well as at least one unique peptide were considered. Identified proteins had to meet the criteria of having at least two peptides, including one unique peptide, to be included in subsequent data analysis. A protein false discovery rate (FDR) of 1% was applied, and it was estimated using the reversed search sequences. Additionally, the maximum number of allowable modifications per peptide was set at 5. The search was performed using a FASTA file containing the *Canis familiaris* proteome downloaded from UniProt as of 14 August 2023.

### 2.5. Statistical Analysis

The MaxQuant ProteinGroups.txt data file was imported into Perseus version 1.6.6.0 [35]. To ensure data integrity, any potential contaminants that did not align with UPS1 proteins were excluded from the dataset. Subsequently, the ion intensities were subjected to a log2 transformation, and comparisons between conditions were performed using *t*-tests. In cases where data were missing, Perseus applied imputation by replacing the values with a constant (zero) to maintain data continuity. For data normalization and the visualization of changes in protein abundance between the control and experimental samples, MetaboAnalyst software version 5.0 was used. The mean central tendency procedure was applied to transform and normalize peptide intensities from the LC-MS analyses. Statistical analysis using analysis of variance (ANOVA) was performed to determine statistically significant proteins (*p* < 0.05) in the data sets. The identified proteins were also submitted to “Stitch EMBL” “http://stitch.embl.de (accessed on 21 August 2023)” to explore their function and understand protein–protein interactions with 5α-dihydrotestosterone (DHT) and finasteride.

## 3. Results

### 3.1. The Prostatic Size before and after Castration

The prostatic volume of dogs diagnosed with BPH from ultrasonography before and after castration is shown in Table 1. All dogs in the BPH group presented decreased prostatic volume resembling the estimated volume one month after castration.

### 3.2. LC-MS/MS Results

The 3850 proteins in all samples were analyzed using in-solution digestion coupled with LC-MS/MS (Appendix A). In heat maps, the data are shown as a grid, with each column indicating a sample and each row indicating a protein (Figure 1A). Partial least squares discrimination analysis (PLS-DA) was used to provide a comprehensive depiction of the data. The PLS-DA plot in two dimensions exhibited discernible clusters among the BPH dogs before castration (BF subgroup) vs. the BPH dogs after castration (AT subgroup) and BF vs. control (CTRL) dogs, while the AT and CTRL dogs appeared to be mostly separate clusters, as illustrated in Figure 1B. Through the utilization of one-way ANOVA with post hoc Tukey’s test, the 22 candidate proteins were identified, including the regulator of G protein signaling 22 (RGS22), apolipoprotein E (APOE), E3 ubiquitin-protein ligase (TRIP12), ubiquitin-associated domain-containing protein 1 (UBAC1), apolipoprotein C-I (APOC1), apolipoprotein A-II (APOA2), solute carrier family 5 member 5 (SLC5A5), mitochondrial ribosomal protein S30 (MRPS30), pseudopodium enriched atypical kinase 1 (PEAK1), tyrosine-protein kinase (TNK2), FRAT regulator of WNT signaling pathway 1 (FRAT1), TAO kinase 1 (TAOK1), probable RNA-binding protein 18 (RBM18), DnaJ homolog subfamily C member 16 (DNAJC16), polymeric immunoglobulin receptor (PIGR), alpha-1-B glycoprotein (A1BG), fetuin B (FETUB), PH domain and leucine rich repeat protein phosphatase 1 (PHLPP1), Ig-like domain-containing protein, neuregulin 1 (NRG1), ring finger protein 213 (RNF213), and voltage-dependent L-type calcium channel subunit alpha (CACNA1D). The assessment of these proteins involved an investigation of their biological processes, cellular components, and molecular functions. This analysis was evaluated using UniProtKB/Swiss-Prot (Table 2). Subsequently, Fisher’s Least Significant Difference Test (Fisher’s LSD) was used to analyze the expression of these 22 proteins across the three groups.

#### 3.2.1. Comparisons of BF Subgroup versus Other Groups (BF vs. AT and BF vs. CTRL)

One unique protein, alpha-1-B glycoprotein (A1BG), demonstrated a significant decrease in the BF subgroup when compared with the other groups (Figure 2A). The overexpression of SLC5A5, TNK2, FRAT1, and PIGR in the BF subgroup compared to the other groups was illustrated using box plots (Figure 2B–E).

#### 3.2.2. Comparisons of AT Subgroup versus Other Groups (BF vs. AT and AT vs. CTRL)

When comparing the AT subgroup to the BF subgroup, a significant decrease was observed in the expression of SLC5A5, TNK2, FRAT1, and PIGR. Moreover, the 12 proteins exhibited highly significant differences between the AT subgroup and CTRL, as well as between the AT subgroup and BF subgroup (Appendix A).

#### 3.2.3. The Interactions between DHT and Overexpressed Proteins in the BF Subgroup

DHT was added to the protein–protein and protein–chemical interaction networks of the SLC5A5, TNK2, FRAT1, and PIGR overexpression in the BF subgroup. Among these, SLC5A5, TNK2, FRAT1, and DHT exhibited the highest level of relationship in the protein interaction network (with edge confidence >0.9) with their functional partners. These partners included cell division cycle 42 (CDC42), growth factor receptor-bound protein 2 (GRB2), glycogen synthase kinase 3 beta (GSK3B), epidermal growth factor receptor (EGFR), breast cancer anti-estrogen resistance 1 (BCAR1), disheveled, Dsh homolog 1 (DVL1), ubiquitin C (UBC), sex hormone-binding globulin (SHBG), androgen receptor (AR), and iodide. However, PIGR was isolated from this protein network (Figure 3).

#### 3.2.4. The Interactions between Finasteride and Overexpressed Proteins in the AT Subgroup

Enrichment assessment and exploration of protein–protein interactions were performed on finasteride and the 12 overexpressed proteins in the AT subgroup, including APOE, APOA2, APOC1, MRPS30, PEAK1, TAOK1, DNAJC16, FETUB, PHLPP1, NRG1, RBM18, and RNF213. Among these proteins combined with finasteride, APOE, APOC1, and APOA2 demonstrated the most significant level of interaction within the protein network (edge confidence >0.9). Additionally, TAOK1, DNAJC16, PHLPP1, and NRG1 displayed strong relationships (edge confidence >0.7) with their respective predicted functional partners. These partners included v-akt murine thymoma viral oncogene homolog 1 (AKT1), Cbl proto-oncogene (CBL), epidermal growth factor receptor (EGFR), v-erb-b2 erythroblastic leukemia viral oncogene homolog 3 (ERBB3), v-erb-a erythroblastic leukemia viral oncogene homolog 4 (ERBB4), forkhead box O1 (FOXO1), growth factor receptor-bound protein 2 (GRB2), heat shock protein 90kDa alpha (cytosolic), class A member 1 (HSP90AA1), heat shock protein 90kDa alpha (cytosolic), class B member 1 (HSP90AB1), lipoprotein receptor-related protein 1 (LRP1), lipoprotein receptor-related protein 8 (LRP8), macrophage stimulating 1 (MST1), mechanistic target of rapamycin (serine/threonine kinase) (MTOR), nitric oxide synthase 3 (NOS3), PH domain and leucine rich repeat protein phosphatase 2 (PHLPP2), RPTOR independent companion of MTOR (RICTOR), SHC (Src homology 2 domain containing) transforming protein 1 (SHC1), sorting nexin family member 27 (SNX27), signal transducer and activator of transcription 3 (STAT3), suppressor of G2 allele of SKP1 (SUGT1). Furthermore, PEAK1 exhibited moderate relationships (edge confidence >0.7) with its predicted functional partner, GRB2. However, FETUB, MRPS30, RNF213, and RBM18 were not connected within this protein network (Figure 4).

## 4. Discussion

To the authors’ knowledge, this is the first investigation to report serum proteome profiling in dogs with BPH before and after castration. A candidate peptide, predicted A1BG, showed lower expression in the BF subgroup compared with other groups. A1BG is a secreted plasma protein whose function is still unknown. Through internal duplication and sequence similarity to immunoglobulin-like proteins, A1BG exhibits homology to the immunoglobulin supergene family. Although this protein is known to be extensively expressed in adult and fetal livers, small amounts of it are discovered in the blood, brain, lung, and lymph nodes [36]. Regarding cancer patients, A1BG level was elevated in the serum of women with endometrial and cervical cancer [37] as well as in the plasma of patients with small renal cell carcinoma [38]. Moreover, the study with plasma proteomic analysis identified A1BG as a potential biomarker related to squamous cell carcinoma of the cervix [39]. In other diseases, increased levels of A1BG have been reported in pediatric steroid-resistant nephrotic syndrome [40] and in dogs with uncomplicated babesiosis [41]. In addition, the A1BG serum biomarker shows potential as a stage III and stage IV endometriosis diagnostic tool for women [42]. Increased thyroid hormone levels in patients with hyperthyroidism might be associated with the regulation of acute phase protein responses, including A1BG, indicating an inflammatory state [43]. A recent study suggested that elephants with negative tuberculosis may have a defense against infection that contains A1BG as part of their possible defensive mechanisms [44]. While canine mammary gland tissue samples, analyzed using matrix-assisted laser desorption/ionization mass spectrometry imaging coupled with liquid chromatography tandem mass spectrometry, exhibited overexpression in poorly differentiated tumors, the lower expression of the A1BG gene was demonstrated to predict a poorer prognosis for distant metastasis-free survival in human breast cancer patients [45]. Therefore, altered A1BG might serve as a potential biomarker for canine prostatic hyperplasia. Further studies are needed to validate A1BG expression and biological function or association with this disease.

Accordingly, in our study, the protein–protein interactions of SLC5A5, TNK2, FRAT1, and DHT were associated with AR, except for PIGR. Functional analysis showed that significantly upregulated proteins related to pathways in cancer, focal adhesion, prostatic cancers, and others were induced as the compensatory reaction. SLC5A5 belongs to the solute carrier (SLC) group of membrane transport proteins, residing within the cell membrane. Encoded by the SLC5A5 gene, the sodium/iodine symporter is an intrinsic membrane protein responsible for facilitating the transfer of iodine from the bloodstream into thyroid follicular cells via SLC5A5 marking the initial stage in the synthesis of thyroid hormone [46]. This transfer leverages the sodium gradient produced by the Na+/K+-ATPase [47]. Previous reports illustrated the induction of the fusion RNA and protein product, namely the SLC45A3-ELK4 transcript, by androgens. Notably, the chimeric mRNA, but not the wild-type ELK4, was identified as a driver for androgen-dependent proliferation in prostate cancer cells. ELK4 protein plays a crucial role in controlling cell overgrowth, operating in both androgen-dependent and -independent prostate cancer cells. Disease progression appears to correlate with the chimeric transcript’s levels, peaking in prostate cancer metastases [48]. A current study in prostate cancer proposed that the abnormal expression of SLC12A5, SLC25A17, and SLC27A6 were strong to metabolic reprogramming and the development of resistance against chemotherapeutic drugs. As prostate cancer cells undergo metabolic reprogramming in an androgen-deprived setting, the upregulation of solute carrier genes points to their potential as attractive therapeutic targets [49]. Moreover, a previous study suggested that SLC5A5 expression could differentiate between follicular adenomas and carcinomas [50]. Our investigation revealed connections between SLC5A5 and the risk of canine BPH. Therefore, the polymorphisms in SLC5A5 might serve as candidates for gene or protein carriers in therapeutic interventions of BPH in dogs. Several investigations have connected various types of cancers to the abnormal activation of tyrosine kinases (TNK2) caused by somatic mutation or DNA amplification [51]. For prostate cancer, tyrosine kinases play an additional role in the progression toward a castration-resistant disease state. This stage of prostate carcinogenesis, which presents the most formidable challenge due to its resistance to effective treatments, is currently without any viable solutions. Non-receptor tyrosine kinases, specifically Src/Etk/FAK collectively known as the Src tyrosine kinase complex, contribute significantly to this process. This complex has been demonstrated to wield considerable influence over the aberrant activation of AR, driven by various growth factors such as EGF, cytokines like IL-6, chemokines including IL-8, and neurokines such as gastrin-releasing peptide. These factors are induced and released from prostate cancer cells to stromal cells upon androgen withdrawal [52]. Elevated signaling through tyrosine kinases has been observed in advanced prostate cancer. Specific tyrosine kinase pathways in the development of prostatic cancer, including the activation of EGFR, ephrin type-A receptor 2, and JAK2, have been identified in a mouse model [53]. As a result, this study suggested that a high level of TNK2 serves as one contributing factor in the development of BPH. The FRAT1 protein functions by inhibiting GSK-3-mediated phosphorylation of beta-catenin, thereby exerting a positive influence on the Wnt signaling pathway. A prior study demonstrated that the expression levels of FRAT1 were modified through overexpression or RNA interference-induced depletion in prostate cancer cells. Notably, FRAT1 was exclusively expressed in the nuclei of normal prostate basal cells, while nuclear FRAT1 was observed in 68% (40 out of 59) of prostate adenocarcinoma samples [54]. Furthermore, instances of FRAT1 overexpression have been documented in various cancers, including ovarian cancer, gastric cancer, esophageal squamous cell carcinoma, and non-small cell lung cancer [55,56,57,58]. The involvement of FRAT1 in cell development and progression is evident. Thus, FRAT1 could potentially contribute as a factor in canine BPH.

Our study demonstrated that the overexpression of PIGR did not show the interaction of PIGR with others such as SLC5A5, TNK2, FRAT1, and DHT. However, in both male and female reproductive organs, androgens have been demonstrated to increase the expression of the PIGR gene, and two crucial AR binding sites have been identified in the human PIGR gene [59]. The high level of PIGR was affected by sex steroids and polypeptide hormones, including estradiol, progesterone, androgens, glucocorticoids, and prolactin [60]. At the basolateral surface of epithelial cells, the PIGR, a member of the immunoglobulin superfamily, binds polymeric immunoglobulin molecules. The complex is subsequently secreted at the apical surface as a secretory component after transcytosis across the cell [61]. One study revealed associations between early-stage endometrial cancer and high PIGR expression, proposing a possible explanation for this less aggressive type [62]. Significant relationships between PIGR expression and low-grade tumors were also discovered in a prospective population-based cohort of epithelial ovarian cancer, demonstrating a less aggressive character for tumors with high PIGR expression [63]. This study demonstrated no association between PIGR and AR or DHT in dogs with BPH, leading them to identify PIGR as a novel biomarker linked to the development of BPH in dogs.

Out of the 11 elevated proteins observed in the AT subgroup when combined with finasteride, apolipoproteins (APOs) exhibited the highest level of prominence within the protein network. APOs play a crucial role in lipid transportation and are essential constituents of lipoproteins, which are intricate structures responsible for ferrying lipids into the bloodstream [64]. These APOs play a crucial role in lipid transport within the blood circulation, contributing significantly to the utilization and clearance of lipoproteins [65]. APOE serves a multifaceted function in cholesterol metabolism, facilitating the uptake of lipoprotein particles into cells by binding to receptors, including those of the low-density lipoprotein receptor family and the receptor for chylomicron remnants [66]. In humans, APOE has been linked not only to cholesterol transport but also to various functions such as DNA synthesis, cell proliferation, angiogenesis, and metastasis. Deviations from these functions can potentially contribute to tumor formation and progression. APOE overexpression has been observed in diverse cancers, including gastric, lung, prostate, thyroid, ovarian, endometrial cancer, and glioblastoma [67,68,69,70,71]. However, APOE plays a role in tissue repair and regeneration beyond its lipid transport. Research indicates that APOE can act as a cell proliferation inhibitor [72], participating in immune regulation, modulation of cell growth and differentiation [73], and exerting antioxidant activity [74]. APOC1, present in both triglyceride-rich and high-density lipoproteins, is critical for plasma lipoprotein metabolism [75]. APOC1’s interactions with APOE are implicated in various biological processes, including cholesterol breakdown, membrane remodeling, and dendritic reorganization [76]. Numerous studies link APOC1 to several diseases, such as diabetic nephropathy, type 1 and type 2 diabetes, Alzheimer’s disease, and glomerulosclerosis [76,77,78,79,80]. Additionally, APOC1’s involvement extends to the progression of breast cancer, pancreatic cancer, lung cancer, and prostate cancer [81,82,83,84]. APOA2, a gene belonging to the apolipoprotein A family, is a primary apolipoprotein found in high-density lipoprotein. The study in pregnant women with gestational diabetes revealed reduced APOA2 expression, indicating its involvement in inflammation [85]. APOA2 has also been linked to cancer and has potential diagnostic and prognostic value in pancreatic cancer, lung cancer, prostatic cancer, colorectal cancer, metastatic renal cancer, and gastric cancer [86,87,88,89,90,91]. In this study, low levels of APOE, APOC1, and APOA2 were observed in the BF subgroup. Recent studies have demonstrated significantly lower levels of serum APOA and APOE in BPH patients compared to controls, while levels were notably higher in BPH than in prostate cancer patients [92]. Similar reductions in serum APOA2 and APOE were observed in dogs with BPH, as in human findings. This study also revealed protein–protein interactions involving APOE, APOC1, and APOA2 in association with finasteride, suggesting the potential effects of this drug on these interactions.

PEAK1 is a novel non-receptor tyrosine kinase that shows widespread expression across all tissues and is highly conserved among vertebrates. Its dysregulation has been observed in various cancer types [93]. Remarkably, PEAK1 exerts significant influence in the realm of cancer. Through the PEAK1-GATA2 transcriptional pathway, it orchestrates the transcription of vascular endothelial growth factor receptor 2 (VEGFR2), a pivotal driver of neovascularization in vertebrates [94]. Recent research has revealed the PEAK1 signature, which becomes elevated under conditions of moderate to severe hypoxia, correlating with heightened MYC expression and a robust Ki67-proliferation index in cancer cells. This phenomenon has been observed in prostate cancer patients [95]. This research observed an association between PEAK1 and finasteride via AKT1 in the protein interaction network. Previous studies have reported that treating rats with finasteride led to a notable reduction in ventral prostate weight and intraprostatic DHT levels, primarily by inhibiting AKT1 and MAPK expression [96]. As a result, elevated PEAK1 levels may potentially be linked to AKT1 and finasteride in dogs.

The discovery of the PHLPP1 stemmed from a search for genes possessing both a phosphatase activity and a PH domain. This gene design was theorized to counteract kinases containing PH domains, such as AKT. The serine/threonine phosphatases PHLPP1 have been established as direct inhibitors of AKT and protein kinase C [97,98,99]. Previous investigations indicated that the absence of PHLPP1, particularly in scenarios involving partial PTEN loss, triggers p53 activation and cellular senescence within the prostate gland. This cascade ultimately contributes to the emergence of spontaneous p53 mutations during the progression of prostate cancer [100]. In this study, the notable elevation of PHLPP1 levels observed in the AT subgroup, relative to the control group, holds promise as a potential marker for BPH in dogs. The protein network showed PHLPP1 associated with AKT1 and finasteride, in which PHLPP1 might be inhibited by AKT1-like finasteride.

Thousand and one kinases (TAOKs) are members of the MAP3K (MAP kinase kinase kinase) family. There are three members of this subfamily known to be found in mammals: TAOK1, 2, and 3 [101]. MAPKs control critical cellular processes including mitosis, proliferation, differentiation, apoptosis, stress, and immune responses [102]. TAOKs are implicated in the regulation of inflammation and immunity. The testes and brain express TAOKs at the greatest levels; however, they are generally ubiquitously expressed in most tissues [103]. The current study demonstrated that TAOK1 increases the lipopolysaccharide (LPS)-induced production of pro-inflammatory cytokines, including IL-6, TNF-α (tumor necrosis factor-α), and IL12p40, in macrophages. It was also found that TAOK1 enhances the LPS-induced activation of ERK1/2 by interacting with TRAF6 (TNF receptor-associated factor 6) and TPL2 (MAP3K8). Therefore, the study proposed that TAOK1 is a positive regulator of the Toll-like receptor 4-induced inflammatory responses in macrophages [104]. Moreover, TAOKs are involved in apoptosis regulation. In the lung carcinoma cell line H1299, it was found that the activation of TAOK1 can induce cell contraction, membrane blebbing, cleavage of Rho kinase 1 and caspase 3, and activate the JNK pathway to induce apoptosis [105]. Our study discovered that TAOK1 is related to AKT1 and finasteride through MST1. Previous reports have indicated that MST1 is also involved in regulating the AKT1 pathway. This kinase could potentially serve as a crucial new connection between androgenic and growth factor signaling, making it a novel therapeutic target in prostate cancer [106]. Therefore, TAOK1 may play a role as one of the factors in managing prostatic diseases.

A large number of chaperones known as heat shock proteins (HSPs) are activated by a variety of stressors, including high temperatures, hypoxia, infections, and other conditions [107]. HSP40 proteins, also known as DNAJ proteins, constitute one of the largest families among HSPs. HSP40 has been linked to cell apoptosis in type 2 diabetes [108,109]. DNAJC is one type of HSP40 protein that only has a J-domain, which is not necessarily located at the N-terminus of the protein [110]. A recent study revealed that ERdj8, a 782-amino-acid protein encoded by DNAJC16, localizes to a meshwork-like endoplasmic reticulum subdomain. It is linked to the control of the size of autophagosomes, together with phosphatidylinositol synthase and the autophagy-related proteins [111]. Moreover, the study in humans suggested that the DNAJC16 gene associated with gout disease may be involved in the mechanism of urate phagocytosis [112]. Interestingly, the DNAJC16 gene is one of the chaperone families HSP40, which regulates AR and mediating sensitivity to chaperone inhibitors to aid in identifying new drug targets for efficacy in castration-resistant prostate cancer [113]. Therefore, this study proposed that DNAJC16 might be a novel therapeutic target in dogs with BPH.

NRG1 is a neurotrophic factor that belongs to the family of epidermal growth factors. It is mainly discovered in the neurological and cardiovascular systems. NRG-1 signaling is transduced by the erb-b2 receptor tyrosine kinase (ErbB) family, which belongs to the receptor protein tyrosine kinase family, and phosphorylation and dimerization of these receptors leads to activation of intracellular signaling cascades [114]. It has been suggested that NRG1/ErbB signaling controls neural development, including neuron–glia interactions during the lamination of the cerebral and cerebellum [115]. In the cardiovascular system, NRG1/ErbBs have systemic effects that include reducing oxidative stress, inhibiting the inflammatory response, protecting central and peripheral nerve performance, and providing comprehensive protection in various clinical situations [116]. Moreover, NRG-1 in cardiomyocytes inhibits apoptosis by PI3K/AKT signaling [117]. NRG-1 has been shown to have anti-inflammatory characteristics in a variety of tissues, including the heart, skin, lungs, brain, and adipose tissue [118]. The mechanisms behind the anti-inflammatory ErbB4 assists result from a variety of processes, such as decreased tissue damage, inhibition of proinflammatory adhesion molecules on endothelial cells, the decreased release of cytokines, and increased macrophage clearance by apoptosis [119]. The study conducted in mice elucidated the functions and interactions of luteinizing hormone and NRG1 in controlling specific testicular processes. These processes include Leydig cell proliferation during testis development through the ERK1/2 pathway, Leydig cell survival in the adult testis via the AKT pathway, and sperm maturation through the maintenance of testosterone production by adult Leydig cells [120]. In the protein network, NRG1 exhibited a similarity to its relation with AKT1 in humans. Therefore, it could potentially be a crucial protein in dogs with BPH.

Furthermore, FETUB, MRPS30, RNF213, and RBM18 did not exhibit any associations in the protein network when combined with finasteride in our study. FETUB is a glycoprotein belonging to the type 3 cysteine protease inhibitor protein family [121]. It is primarily produced by the liver and inhibits the activation of cysteine-type endopeptidases. FETUB shares some similarities in function with its paralog fetuin-A, which plays roles in various physiological processes such as fatty acid transport [122], response to systemic inflammation [123], and the inhibition of basic calcium phosphate precipitation [124]. Recent studies in humans have linked FETUB to cardiovascular disease [125], tumors [126], reproduction [127], and glucose and lipid metabolism [128]. Interestingly, FETUB expression in rats is induced by estrogen and may inhibit breast cancer development under estrogen influence [129]. In humans, FETUB levels were significantly higher in normal prostate cell lines compared to prostate cancer (PC) cells, and FETUB overexpression inhibited PC cell proliferation, migration, and invasion by inhibiting the PI3K/AKT signaling pathway and inducing apoptosis [130]. Mitochondrial ribosomal protein (MRP) genes, including MRPS30, have been linked to human diseases such as deafness and retinitis pigmentosa [131]. MRPS30, an apoptosis-related gene, exhibited differential expression between androgen-resistant and androgen-responsive prostate cancer cell lines [132]. Recent research has proposed hypotheses about the MRPS30 genomic region and its relation to postmenopausal breast cancer risk [133]. However, its precise function remains elusive. In Moyamoya disease, a rare cerebrovascular condition, RNF213 mutations are common [134]. This gene encodes a protein with AAA+ ATPase and E3 ubiquitin ligase domains, which are associated with functions like angiogenesis, autophagy, autoimmunity, and lipid metabolism [135]. RNF213 has been identified as an interferon-induced megaprotein with antimicrobial activity against various viruses [136]. In Madin Darby canine kidney cells, RNF213 genes were upregulated when exposed to acid in a metabolic acidosis model, suggesting a potential role as a ubiquitin ligase [137]. Taken together, these findings suggest that FETUB, MRPS30, and RNF213 may be novel proteins with potential relevance to canine BPH. RNA-binding proteins (RBPs) facilitate the interactions among various RNAs, forming ribonucleoproteins that govern post-transcriptional gene expression regulation [138]. Moreover, they play a vital role in regulating RNA metabolism processes like splicing, translation, and localization [139]. Given the regulatory functions of RBPs, recent research has pointed about their malfunction in cancer initiation and progression [140]. In this study, RBM18 exhibited elevated levels in the AT subgroup. However, there is no prior documentation of RBM18 in mammals, suggesting it could be a novel protein associated with the development of BPH in dogs.

## 5. Conclusions

This study performed the first investigation of serum proteome profiling in dogs with BPH before and after castration. Interestingly, the predicted A1BG displayed lower expression in the BF subgroup when compared to the other groups. Additionally, our study unveiled protein–protein interactions involving SLC5A5, TNK2, FRAT1, and DHT with AR. Furthermore, within the AT subgroup, where finasteride was combined, this study observed an elevation in 12 proteins, with a particular prominence of APOs. These findings indicate the potential of these proteins as biomarkers and warrant further investigation into their roles and implications in canine BPH. However, further studies are essential to validate their expression, elucidate their biological functions, and establish their associations with this disease.

## Figures and Tables

**Figure 1 animals-13-03853-f001:**
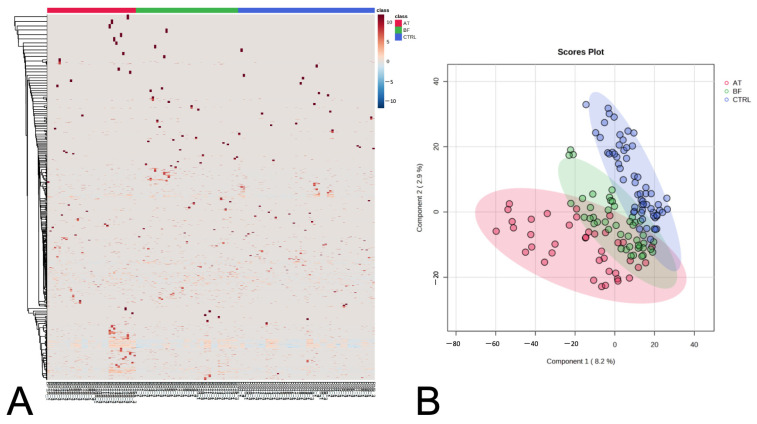
Heat map of identified serum proteins, with each column indicating a sample and each row indicating a serum protein (**A**). The PLS-DA scores plot of components one and two, comparing serum samples as they cluster by sampling group (**B**).

**Figure 2 animals-13-03853-f002:**
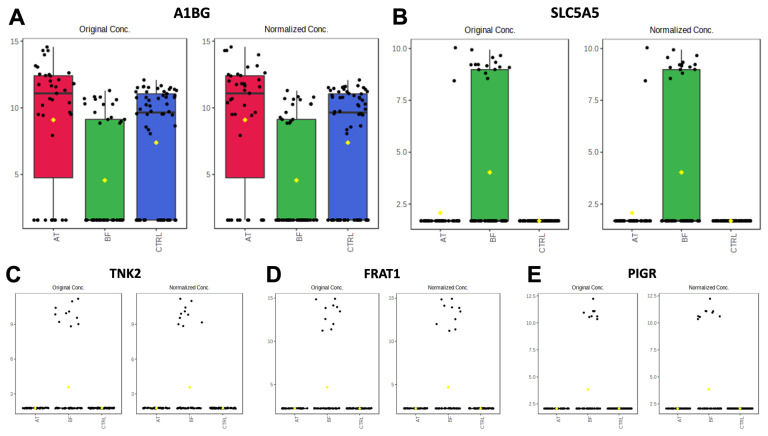
Boxplot of protein expression significantly different between AT (red), BF (green), and CTRL (blue) of alpha-1-B glycoprotein (**A**), solute carrier family 5 member 5 (**B**), tyrosine-protein kinase (**C**), FRAT regulator of WNT signaling pathway 1 (**D**) and polymeric immunoglobulin receptor (**E**). The distribution of differences is shown as a box plot. *y*-axis: intensity of mass spectrum; *x*-axis: the different experimental groups.

**Figure 3 animals-13-03853-f003:**
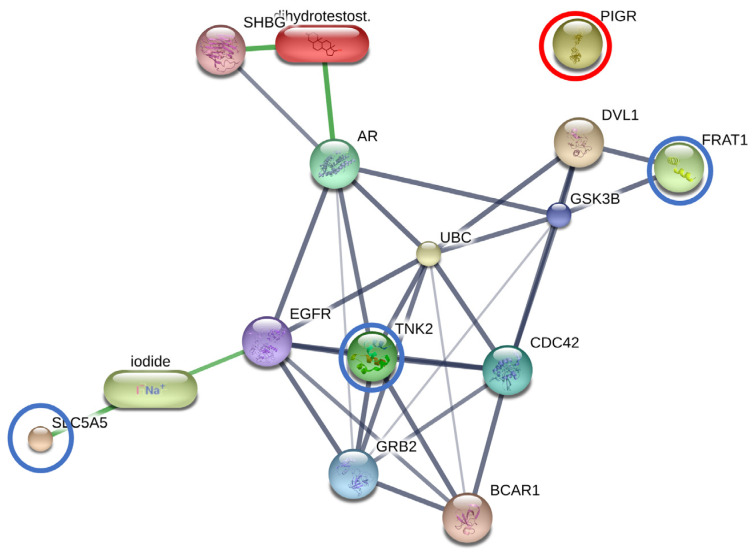
Blue circles: involvement of solute carrier family 5 member 5 (SLC5A5), tyrosine-protein kinase (TNK2), and FRAT regulator of WNT signaling pathway 1 (FRAT1) in the network of protein interaction with dihydrotestosterone (DHT) their functional partners. Red circle: polymeric immunoglobulin receptor (PIGR). Abbreviations of functional protein partner: CDC42, cell division cycle 42; GRB2, growth factor receptor-bound protein 2; GSK3B, glycogen synthase kinase 3 beta; EGFR, epidermal growth factor receptor; BCAR1, breast cancer anti-estrogen resistance 1; DVL1, disheveled, Dsh homolog 1; UBC, ubiquitin C; SHBG, sex hormone-binding globulin; AR, androgen receptor.

**Figure 4 animals-13-03853-f004:**
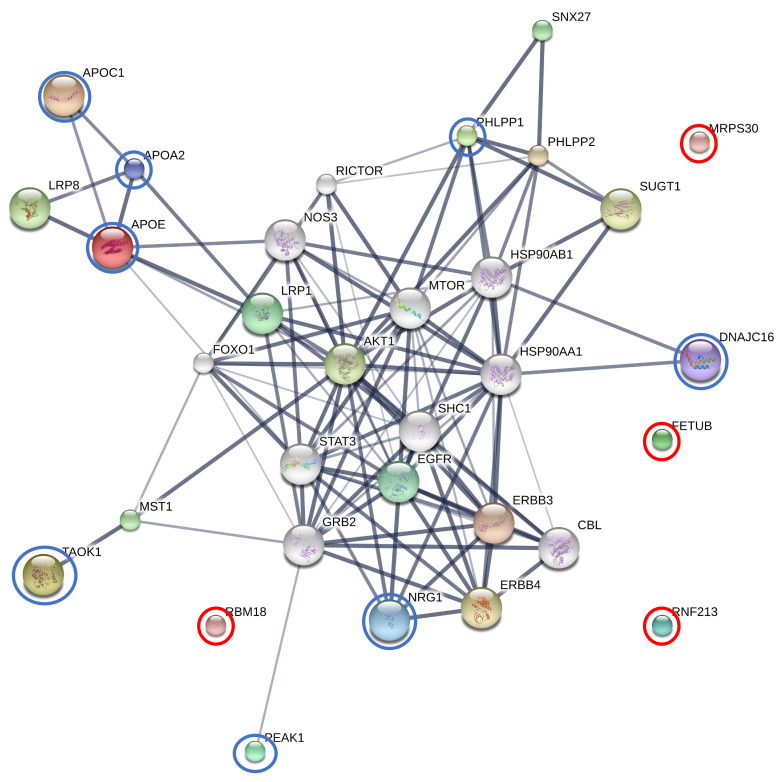
Blue circles: involvement of apolipoprotein C-I (APOC1), apolipoprotein E (APOE), apolipoprotein A-II (APOA2), TAO kinase 1 (TAOK1), DnaJ homolog subfamily C member 16 (DNAJC16), PH domain and leucine rich repeat protein phosphatase 1 (PHLPP1), Ig-like domain-containing protein, neuregulin 1 (NRG1) and pseudopodium-enriched atypical kinase 1 (PEAK1) in the network of protein interaction with finasteride and their respective predicted functional partners. Red circle: mitochondrial ribosomal protein S30 (MRPS30), fetuin B (FETUB), ring finger protein 213 (RNF213) and RNA-binding protein 18 (RBM18). Abbreviations of functional protein partner: AKT1, v-akt murine thymoma viral oncogene homolog 1; CBL, Cbl proto-oncogene; EGFR, epidermal growth factor receptor; ERBB3, v-erb-b2 erythroblastic leukemia viral oncogene homolog 3; ERBB4, v-erb-a erythroblastic leukemia viral oncogene homolog 4; FOXO1, forkhead box O1; GRB2, growth factor receptor-bound protein 2; HSP90AA1, heat shock protein 90kDa alpha (cytosolic), class A member 1; HSP90AB1, heat shock protein 90kDa alpha (cytosolic), class B member 1; LRP1, lipoprotein receptor-related protein 1; LRP8, lipoprotein receptor-related protein 8; MST1, macrophage stimulating 1; MTOR, mechanistic target of rapamycin (serine/threonine kinase); NOS3, nitric oxide synthase 3; PHLPP2, PH domain and leucine rich repeat protein phosphatase 2; RICTOR, RPTOR independent companion of MTOR; SHC1, SHC (Src homology 2 domain containing) transforming protein 1; SNX27, sorting nexin family member 27; STAT3, signal transducer and activator of transcription 3; SUGT1, suppressor of G2 allele of SKP1.

**Table 1 animals-13-03853-t001:** Age, weight, breed, and ultrasonographic findings in 15 dogs diagnosed with benign prostatic hyperplasia.

Dog	Breed	Weight	Age	Estimated Volume	Prostatic Volume before Castration	Prostatic Volume after Castration
		(kg)	(Years)	(cm^3^)	(cm^3^)	(cm^3^)
1	American Bully	30.00	2	13.18	33.80	7.70
2	Boston Terrier	12.00	5	7.24	9.12	7.20
3	Chihuahua	3.20	7	4.34	6.80	4.37
4	Chihuahua	2.60	3	4.14	5.80	3.20
5	Chihuahua	4.90	6	4.90	7.20	3.56
6	Golden Retriever	45.00	8	18.13	24.70	9.47
7	Jack Russell Terrier	8.50	7	6.09	7.80	3.60
8	Mongrel	24.00	7	11.20	43.60	9.30
9	Pomeranian	4.00	6	4.60	6.30	4.30
10	Pomeranian	4.30	10	4.69	9.04	4.11
11	Pomeranian	7.30	6	5.69	10.60	3.66
12	Shih Tzu	5.70	7	5.16	5.35	2.57
13	Shih Tzu	6.90	4	5.56	8.97	4.50
14	Shih Tzu	8.50	10	6.09	7.60	4.10
15	Thai Bangkaew	16.50	4	8.73	9.09	3.00

**Table 2 animals-13-03853-t002:** Nominated proteins based on biological process, cellular components, and molecular functions involvement using UniProtKB/Swiss-Prot.

Protein Names	Gene Names	−log10(p)	FDR	Fisher’s LSD	Biological Process	Cellular Component	Molecular Function
Regulator of G protein signaling 22	RGS22	14.802	6.07 × 10^−12^	AT—BF; AT—CTRL; BF—CTRL	Regulation of signal transduction	N/A	G-protein alpha-subunit binding
Apolipoprotein E	APOE	9.3962	7.73 × 10^−7^	AT—BF; AT—CTRL	AMPA glutamate receptor clustering, cGMP-mediated signaling, cholesterol catabolic process, G protein-coupled receptor signaling pathway, gene expression	Chylomicron, endoplasmic reticulum, extracellular exosome, extracellular matrix, extracellular space, Golgi apparatus, plasma membrane	Amyloid-beta binding, antioxidant activity, cholesterol transfer activity, enzyme binding, heparin binding, lipoprotein particle binding, phospholipid binding
E3 ubiquitin-protein ligase	TRIP12	8.341	5.85 × 10^−6^	AT—BF; AT—CTRL; BF—CTRL	DNA repair, protein ubiquitination, ubiquitin-dependent protein catabolic process	Nucleoplasm	Ubiquitin protein ligase activity, zinc ion binding
Ubiquitin-associated domain-containing protein 1	UBAC1	8.1622	6.62 × 10^−6^	AT—BF; AT—CTRL; BF—CTRL	N/A	N/A	N/A
Apolipoprotein C-I	APOC1	6.7248	1.31 × 10^−4^	AT—BF; AT—CTRL	Lipid transport, lipoprotein metabolic process, negative regulation of cholesterol transport, triglyceride metabolic process	High-density lipoprotein particle, very-low-density lipoprotein particle	Fatty acid binding, phospholipase inhibitor activity
Apolipoprotein A-II	APOA2	6.6902	1.31 × 10^−4^	AT—BF; AT—CTRL	Lipid transport, lipoprotein metabolic process	Extracellular region	Lipid binding
Solute carrier family 5 member 5	SLC5A5	6.3569	2.42 × 10^−4^	BF—AT; BF—CTRL	N/A	Membrane	Transmembrane transporter activity
Mitochondrial ribosomal protein S30	MRPS30	5.6704	0.001028	AT—BF; AT—CTRL	Translation	Mitochondrial large ribosomal subunit	Structural constituent of ribosome
Pseudopodium enriched atypical kinase 1	PEAK1	5.6141	0.0010402	AT—BF; AT—CTRL	Protein phosphorylation	N/A	ATP binding, protein kinase activity
Tyrosine-protein kinase	TNK2	5.4357	0.001384	BF—AT; BF—CTRL	Adaptive immune response, intracellular signal transduction, protein phosphorylation	Plasma membrane	ATP binding, metal ion binding protein, tyrosine kinase activity
FRAT regulator of WNT signaling pathway 1	FRAT1	5.4029	0.001384	BF—AT; BF—CTRL	N/A	N/A	N/A
TAO kinase 1	TAOK1	5.2529	0.0017921	AT—BF; AT—CTRL	Protein phosphorylation	N/A	ATP binding, protein kinase activity
RNA-binding protein 18	RBM18	4.9606	0.0032425	AT—BF; AT—CTRL	N/A	N/A	RNA binding
DnaJ homolog subfamily C member 16	DNAJC16	4.8531	0.0037381	AT—BF; AT—CTRL	N/A	Membrane	N/A
Polymeric immunoglobulin receptor	PIGR	4.8367	0.0037381	BF—AT; BF—CTRL	N/A	Membrane	N/A
Alpha-1-B glycoprotein	A1BG	4.6873	0.0049434	AT—BF; CTRL—BF	N/A	Membrane	N/A
Fetuin B	FETUB	4.4265	0.008483	AT—BF; AT—CTRL	Binding of sperm to zona pellucida, negative regulation of endopeptidase activity	Extracellular space	Cysteine-type endopeptidase inhibitor activity, metalloendopeptidase inhibitor activity
PH domain and leucine rich repeat protein phosphatase 1	PHLPP1	4.263	0.011673	AT—BF; AT—CTRL	N/A	N/A	N/A
Ig-like domain-containing protein	N/A	4.2288	0.011965	AT—CTRL; BF—CTRL	N/A	N/A	N/A
Neuregulin 1	NRG1	4.1774	0.012795	AT—BF; AT—CTRL	Nervous system development	Cellular anatomical entity	Signaling receptor binding
Ring finger protein 213	RNF213	4.1374	0.013005	AT—BF; AT—CTRL	N/A	Cytoplasm	ATP hydrolysis activity, metal ion binding, ubiquitin-protein transferase activity
Voltage-dependent L-type calcium channel subunit alpha	CACNA1D	4.1289	0.013005	AT—BF; AT—CTRL; CTRL—BF	Regulation of monoatomic ion transmembrane transport	Voltage-gated calcium channel complex	Metal ion binding, voltage-gated calcium channel activity

## Data Availability

The MS/MS raw data and analysis files have been deposited in the ProteomeXchange Consortium “http://proteomecentral.proteomexchange.org (accessed on 11 October 2023)” via the jPOST partner repository (https://jpostdb.org) with the data set identifier JPST002343 and PXD046059 (preview URL for reviewers: https://repository.jpostdb.org/preview/62420013365269e3ad246e, Access key: 8273).

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
