# Peer review of "Comparative Serum Proteome Profiling of Canine Benign Prostatic Hyperplasia before and after Castration"

_animals, 2023, doi:10.3390/ani13243853_

Round 1

Reviewer 1 Report

Comments and Suggestions for Authors

This paper presents interesting data on the proteome profile in dogs with benign prostatic hyperplasia (BPH), offering potential value for future clinical veterinary diagnostics. However, a major methodological concern arises from the presented in M&M clinical diagnosis of BPH in the studied dogs. According to the Materials and Methods section, the diagnosis was based solely on the evaluation of the size of the prostate gland, with glands of higher volume being classified as BPH.

This methodology poses a significant risk of including dogs in the study who may be suffering from other prostatic disorders leading to volume increase, but are not actually cases of BPH. This potential inclusion of unrelated cases could introduce bias to the obtained results. Detailed comments on this issue are provided below:

Line 60: result in estrogen metabolites – please re-write, what are the estrogen metabolites and where are they produced?

Line 61: linked to free radicals is a bit too general, please be more specific

Line 61: prolactin in male dog where does it come from?

Line 65: currently US is the main diagnostic tool to evaluate prostate parenchyma, please change the order / wording to state it

Line 66: authors mention FNA but in 73 biopsy – are these the same or 2 different procedures

Line 72 – 75: there is one more risk with FNA/biopsy – seeding neoplastic cells (or bacteria) with the needle – please add

Line 128: the ultrasonographic evaluation should also include assessment of echostructure of the gland, echogenity, homegenity, heterogeneity, symmetry and parenchymal anechoic structures presence and appearance – please consider and include information how the prostate parenchyma was assessed in your study.

Line 134: why D30 was chosen, is there a chance that the prostatic changes after castration may prevail longer, same with DHT related metabolites in serum?

Line 207: was the BPH diagnosed only based on the prostatic volume? How were other prostatic disorders ruled out?

Reviewer 2 Report

Comments and Suggestions for Authors

In this article, Ploypetch et al aimed to characterize the proteome of dogs with benign prostatic hyperplasia before and after castration and also as compared to healthy dogs. This approach is of utmost importance not only for delineating the associated pathways but also to identify potential biomarkers for clinical diagnosis.

-The Introduction section has a thorough analysis of the related literature.

-The Materials & Methods section has the appropriate level of detail.

-The differential expression of proteins between the groups is not clear. There should be a distinction between differentially expressed proteins for i. control vs BF, ii. BF vs AT but also iii. control vs AF groups. In this way the associated pathways will be better understood and exploited for further analyses. The addition of subsections in the Results section that discuss each comparison individually would help. 

-In Figure 2 and suppl. figures, what is the legend of the y-axis? What is protein expression normalized to?

-The Discussion section presents a thorough analysis of each identified protein.

Round 2

Reviewer 1 Report

Comments and Suggestions for Authors

Authors answered all my comments

Reviewer 2 Report

Comments and Suggestions for Authors

I would like to thank the authors for their work and additions to the manuscript. The comparisons are now clear and the manuscript is suitable for publication.